# Modulation of CD4 T Cell Response According to Tumor Cytokine Microenvironment

**DOI:** 10.3390/cancers13030373

**Published:** 2021-01-20

**Authors:** Théo Accogli, Mélanie Bruchard, Frédérique Végran

**Affiliations:** 1Faculté des Sciences de Santé, Université Bourgogne Franche-Comté, 21000 Dijon, France; theo.accogli@inserm.fr (T.A.); melanie.bruchard@u-bourgogne.fr (M.B.); 2Team “CAdIR”, CRI INSERM UMR1231 “Lipids, Nutrition and Cancer”, Dijon 21000, France; 3LipSTIC LabEx, 21000 Dijon, France; 4Centre Georges François Leclerc, 21000 Dijon, France

**Keywords:** CD4, cytokines, cancer, immunotherapy

## Abstract

**Simple Summary:**

It is now accepted that CD4 T lymphocytes play an essential role in the anti-tumor response. CD4 T lymphocytes can activate and regulate several aspects of innate and adaptive immunity and participate in the rejection of tumors. Understanding the impact of the tumor, through cytokines present in the microenvironment, but also the effect of anti-cancer therapies are critical aspects of immunotherapy research aiming at improving the anti-tumor response dependent on CD4 T lymphocytes.

**Abstract:**

The advancement of knowledge on tumor biology over the past decades has demonstrated a close link between tumor cells and cells of the immune system. In this context, cytokines have a major role because they act as intermediaries in the communication into the tumor bed. Cytokines play an important role in the homeostasis of innate and adaptive immunity. In particular, they participate in the differentiation of CD4 T lymphocytes. These cells play essential functions in the anti-tumor immune response but can also be corrupted by tumors. The differentiation of naïve CD4 T cells depends on the cytokine environment in which they are activated. Additionally, at the tumor site, their activity can also be modulated according to the cytokines of the tumor microenvironment. Thus, polarized CD4 T lymphocytes can see their phenotype evolve, demonstrating functional plasticity. Knowledge of the impact of these cytokines on the functions of CD4 T cells is currently a source of innovation, for therapeutic purposes. In this review, we discuss the impact of the major cytokines present in tumors on CD4 T cells. In addition, we summarize the main therapeutic strategies that can modulate the CD4 response through their impact on cytokine production.

## 1. Introduction

CD4 T cells are key components of the immune system that shape the anticancer immune response in animal models but also in humans. Studies have shown that CD8 response requires CD4 “helper” functions [1,2,3]. At the same time, CD4 T cells can have cytotoxic activity and during adoptive transfer experiments in mice, they can induce an anti-tumor response [4]. Another study revealed that CD4 T cells can recognize neoantigens in melanoma [5]. In addition, patients who have received neoantigenic vaccines show induction of an antigen-specific CD4 T response [6,7]. The biology of CD4 T cells is complex because naive T cells can differentiate into various subpopulations with different functions. Since the original classification by Mosmann and Coffman of CD4 T lymphocytes into Th1 (Thelper 1) and Th2 subsets [8], the repertoire of CD4 T cell subsets has expanded to include additional effector T cell subsets like Th17 cells [9,10,11], Follicular helper T cells (Tfh) [12] and Foxp3 (Forkhead box P3) regulatory T cells(Tregs) [13]. The relevance of these CD4 T cell subtypes has been documented in various immunopathological conditions such as inflammatory diseases and cancer [14,15,16]. In 2008, IL (Interleukin)-9-producing CD4 T helper cells (Th9) were identified as a new subset of CD4 T helper cells that are proinflammatory in vivo [17,18]. IL-2 and IFNγ (Interferon) producing CD4 T cells such as Th1 have been shown to play an essential role in the induction and persistence of anti-tumor CD8 T cells. Conversely, Tregs have a pro-tumor role through their immunosuppressive activity. Th2 and Th17 are thought to have pro-tumor functions due to the cytokines they produce [19,20] but their roles are still controversial. Understanding the differentiation mechanisms of these cells, but also the impact of the tumor environment on their differentiation and activity could allow the development of innovative therapeutic strategies.

## 2. Cytokine Environment of Tumors

Cytokines are chemical mediators synthesized by different cells in an organism. They have been grouped into subsets according to the cells that produce them (monokines produced by myeloid cells, lymphokines produced by lymphocytes, interleukins produced by leukocytes, and chemokines responsible for the migration of leukocytes). They can also be classified according to their predominant role: interleukins, interferons, TNFs (Tumor Necrosis Factor), growth factors, and chemokines. Interleukins are cytokines that serve as a messenger between different immune cells. They are mainly produced during inflammatory reactions. Tumors are indeed composed of multiple cell types, including innate and adaptive immune cells that have a profound impact on tumor progression [21]. However, immune activation during tumor progression can lead to contrasting outcomes. Chronic activation of immune effector cells by tumor cells triggers the release of pro-inflammatory cytokines, chemokines, and other bioactive mediators, thereby creating an inflammatory milieu. This chronic inflammation, which has been observed in cases of obesity or smoking, leads to high risks of cancer [22]. In addition, recent studies show that unresolved inflammation promotes tumor progression, for example by promoting the stability of p53 mutants [23]. In contrast, the immuno-surveillance concept, initially brought up by Burnet in 1957 and later substantiated by mouse and human findings, states the immune system is capable of limiting tumor development [24]. In this context, there is a balance shift from inhibitory to activating cytokines to generate a protective anti-cancer response.

TGFβ (Transforming Growth Factor) is frequently present in the tumor microenvironment. It was initially described as a signal to prevent premalignant progression but became a factor that malignant cells use to their own advantage. The presence of these immune-infiltrating cells, known for their capacity to produce TGFβ coincides with the amount of TGFβ [25] providing additional complexity in the regulation of this cytokine.

Tissue damage occurring during tumor development and the release of alarm cytokines such as IL-1α, IL-1β, and TNFα induce the production of IL-6, IL-10, IL-11, and IL-23 by tissues and myeloid cells. These cytokines, which allow an autoregulation loop under homeostatic conditions, will promote the proliferation of tumor cells, the healing of damaged tissues but also angiogenesis and tumor vascularization by induction of VEGF (Vascular Endothelial Growth factor) [26].

The inflammasome is a protein complex originally described in innate immune cells such as macrophages. The inflammasome sets up after the detection of so-called danger signals. A first signal (priming), generally dependent on TLR (Toll-Like Receptor) will allow the induction of NLRP3 (NOD-Like Receptor family Pyrin domain containing 3) and IL-1β gene expression [27]. A second signal, such as efflux of potassium following an alteration of the cell membrane, lysosomal destabilization, or even mitochondrial damage, activates the inflammasome and induces its formation. The composition of the inflammasome varies depending on the activator. The NLRP3 inflammasome is the best described. It is made up of the NLRP3 protein, ASC (Apoptosis-associated Speck-like protein containing a CARD) adapter, and Caspase-1, which is activated when the inflammasome sets in. The Caspase-1 once activated can participate in the maturation of the pro-inflammatory cytokines IL-1β and IL-18 [28]. Some cancer, including advanced melanoma, spontaneously release IL-1β [29]. IL-33 is a member of the IL-1 superfamily [30]. It is now well known to have an important role in innate and adaptive immunity through its contribution to tissue homeostasis and to stress responses such as tumor development. IL-33 is constitutively expressed at high levels in the nucleus of human and murine tissue mucosa and in various cell types, such as endothelial cells [31,32] and barrier tissue epithelial cells [32,33]. Fibroblast reticular cells of lymphoid tissues and cells of the central nervous systems represent a major source of IL-33 [32,34].

Immune cells are the main cytokine producers in tumors. IL-2 is produced mainly by CD4 T cells simulated by an antigen, but also by NK (Natural Killer) and activated dendritic cells [35,36]. TNFα is produced primarily by activated macrophages, but can also be produced by many other immune cells. IL-4, the most important Th2 cytokine, is mainly produced by activated T cells, mast cells, basophils, and eosinophils. IFNγ is predominantly produced by NK cells in the innate arm of the immune system, as well as by effector T cells like CD4 and CD8 T cells in the adaptive arm of the immune system and could be regulated by miRNA [37,38]. Innate lymphoid cells have also been shown to produce IFNγ [39,40]. Varying degrees of IFNγ are also secreted by mucosal epithelial cells and macrophages [41]. Regarding the cell source of IL-27, this cytokine is mainly produced by cells of myeloid origin such as monocytes, macrophages, dendritic cells, and microglial cells, in response to stimuli acting through Toll-like receptors [42] or TNF-R-family members, for example, CD40L [43]. Finally, IL-12 was originally identified as an NK cell-stimulatory factor with multiple biologic effects on lymphocytes [44]. It is mainly produced by APCs (Antigen Presenting Cells) and B cells upon TLR engagement [45].

## 3. Impact of Cytokines from the Tumor Microenvironment on CD4 T Cells

### 3.1. Protumor Effects

#### 3.1.1. TGF β

TGFβ is a pleiotropic cytokine that acts by the engagement of a tetrameric receptor composed of the TGFβI and II receptors. The binding of TGFβ to its receptor triggers serine/threonine kinase receptor activity, allows phosphorylation of downstream signaling targets. TGFβ signaling is primarily mediated by the SMAD family transcription factors but is also known to initiate Smad-independent pathways. The induction of Tregs peripheral differentiation is directed, among other things, by environmental stimuli such as the presence of TGFβ [46]. The key role of TGFβ in inducing Foxp3 expression in Tregs was identified by studies showing that deletion of CNS1 (Conserved Non-coding Sequence) region of Foxp3 (which contains the conserved Smad3 binding sequence) or the Smad binding site itself, results in a reduction of Tregs [47]. Moreover, the TGFβ present in the tumor microenvironment can interfere with conventional T cell activation by inhibiting TCR (T Cell Receptor) signaling [48,49]. It is also able to inhibit T helper subtypes by suppressing the expression of the transcription factors defining their profile, such as Tbet (T-box transcription factor) and Gata3, which are essential for Th1 and Th2 respectively [50,51]. Several works have demonstrated the redundant roles of the Smad2 and Smad3 transcription factors in the inhibition of Th1 differentiation by TGFβ [52]. The Smad signaling pathway has also been shown to induce the transcription factor SOX4 (SRY-Box Transcription Factor) which interferes with Gata3 activity during Th2 differentiation [53]. A recent study showed that the specific targeting of TGFβ signaling in CD4 T cells, via a TGFBR2 coupled to a CD4 antibody induces an anti-tumor response mediated by IL-4 in a mouse breast cancer model [54]. Finally, Th17 cells harbor ambivalent effect on cancer [55] but it seems that TGFβ could be responsible for immunosuppressive and pro-tumor functions of Th17 cells [56] (Table 1).

#### 3.1.2. IL-6

Canonical IL-6 signaling pathway depends on its IL-6R membrane receptor, which is a common receptor for IL-6, IL-11, LIF (Leukemia Inhibitory Factor), Oncostatin, cariotrophine-1, and IL-35. The homodimerization of this receptor during IL-6 binding leads to the activation of Tyrosine Kinases associated with IL-6R, JAK1 (Janus Kinase), JAK2, and TYK2 (Tyrosine Kinase) and to the phosphorylation, translocation, and activation of STAT1 (Signal Transducer and Activator of Transcription) and STAT3 [97,98]. Induction of this pathway is responsible for T cell proliferation and commitment to the Th17 and Tfh subtypes [99,100]. IL-6 blocks Foxp3 activity, limiting the ability of TGFβ to promote the development of Tregs. Thus, by participating in the Th17 differentiation, IL-6 amplifies the pro-inflammatory response [57,58,101] (Figure 1 and Table 1).

#### 3.1.3. IL-2

IL-2 is a key regulator of T cell survival, differentiation, and proliferation [102]. It is the most studied cytokine in the field of cancer immunotherapy [103]. Activation of the CD25 receptor (or IL-2R) by IL-2 will lead to cytosolic signaling pathways dependent on serine/threonine kinases which make it possible to initiate the activity of mTORC1 (Target Of Rapamycin Complex), which in turn modulates cell metabolism [104]. However, STAT5 is the critical mediator of IL-2 signals [105,106,107]. Analysis of STAT5 binding sites using Chromatin Immunoprecipitation in CD4 T cells have shown that a number of essential transcription factors defining CD4 T cells are direct STAT5 target genes. These include Foxp3 [59,60], Tbet21 [61], Maf [62], and Gata3 [63]. However, not all STAT5 binding sites are associated with increases in gene expression: binding of STAT5 to the *il17a* locus is repressive, thus inhibiting STAT3-mediated transcription of the *il17* gene and suppressing the differentiation into Th17 cells [64]. Likewise, STAT5 has been shown to bind to the *bcl6* promoter in CD4 T cells [65,108], correlating with repressed expression of the gene and thus limiting the differentiation of TFh [65]. The IL-2 receptor (IL-2R) is made up of 3 subunits α,β,γ. IL-2 has a better binding affinity with the IL-2Rαβγ complex than with the other combinations and in particular the simple or heterodimeric IL-2R [109]. Tregs have the greatest affinity for IL-2 among T cells [102]. These cells, which prevent the development of autoimmunity under homeostatic conditions [110] have immunosuppressive and pro-tumor effects. CD4 T lymphocytes evolving in a tumor environment rich in IL-2 (in association with TGFβ) express the transcription factor Foxp3, which impairs the differentiation of Tregs and the production of IL-10, that participate in an immunosuppressive microenvironment [66,67,68]. In this context, Tregs are capable of suppressing the anti-tumor functions of CD4, CD8, and NK cells, leading to an absence of effective anti-tumor immune response (Figure 1 and Table 1).

#### 3.1.4. IL-1

IL-1β is undoubtedly a cytokine with an ambivalent role. Indeed, the IL-1β secreted in the tumor microenvironment induces the secretion of IL-17 through its effect on CD4 T cells, by driving the differentiation and expansion of Th17 cells, which promote angiogenesis and tumor growth via the STAT3 signaling pathway [69,80]. IL-17 also recruits MDSCs, exhibiting major immunosuppressive activity [70,71]. Conversely, for several years, studies have shown that IL-1β can participate in the eradication of tumors. Indeed, it is one of the cytokines constantly associated with the effective eradication of cancer by a Th1 response, in particular in myeloma and B lymphoma [72]. In this study, the authors propose that IL-1α and β could promote tumor progression, but in association with other cytokines, in particular, those signaling a Th1 response and their effects on macrophages, they rather promote an anti-tumor immune response. In another study, a team confirmed that IL-1α and β are essential for a complete Th1- induced anti-tumor response in melanoma [73]. Th9 cells are closely related to the Th2 lineage, which generally antagonizes Th1 responses. However, Th9 cells and their anti-tumor cytokine IL-9 are powerful anti-tumor agents, which can be exploited in cell therapy [111]. Typically induced by IL-4 and TGFβ, the absence of the latter could be replaced by IL-1β to promote Th9 cells [112]. Moreover, Th9 cells differentiated in presence of TGFβ and IL-4 harbor higher anti-tumor activity when IL-1β is present [74] (Table 1). In short, it seems that the concentration of IL-1 is the key to understand its effects on tumor growth. While a low concentration would trigger pro-tumor events in the tumor microenvironment and stimulate tumor growth, metastasis, and angiogenesis, high doses of IL-1 promote anti-tumor responses. However, it is necessary to keep in mind that high concentrations of IL-1 can have serious toxic effects [113] (Figure 1).

#### 3.1.5. TFNα

TNFα is a powerful anti-tumor cytokine. TNFα was identified in 1975 and named after its ability to induce necrosis of mice sarcomas when injected at high concentrations [75]. Signaling cascades induced by its binding to one of its receptors (TNFR1, TNFR2) induce cell death by necrosis or apoptosis. This is why TFNα was one of the first cytokines clinically used for the treatment of cancer [114]. Concerning T cells, TNFα appears to be one of the weapons of antigen-specific CD8 T cells to eradicate tumor cells [76]. To focus on CD4, a recent study used CD4 T cell-based adoptive immunotherapy to treat CT26 colorectal tumors. The authors show that antigen-specific CD4 T cells can eradicate established CT26 tumors when injected after cyclophosphamide treatment. This is achieved through cytokine-mediated CD4 response, and mostly via TNFα, which synergize with chemotherapy to induce ROS (Reactive Oxygen Species) in CT26 cells [115]. The authors also mention that IFNγ may be necessary and that the efficacy of T cells to achieve a complete eradication may rely on the concomitant secretion of both IFNγ and TNFα. Still, the anti-tumor effects of TNFα require greater concentrations than cells can physiologically produce within the tumor. Unexpectedly from a cytokine called after its ability to induce cancer cell death, pro-tumor effects of TNFα have also been reported. Indeed, TNFα deficient mice as well as TNFR1 or TNFR2 deficient mice are resistant to skin tumor development [116]. The mechanisms underlying this may imply CD4 T cells. Indeed, although considered to be an anti-tumor factor at high concentrations [117], TNFα exhibits a deleterious effect due to its ability to stimulate regulatory CD4 T cells. Several studies carried out in vitro using mouse material show an increase in the proliferation, survival, stability, expression of CD25 and Foxp3, as well as an increase in the immunosuppressive functions of Tregs, treated with TNFα [77,78,118,119] (Table 1). Tregs from human blood all express high levels of TNFR2 while other CD4 T cells express lower levels of this receptor [120]. Likewise, in mice, more than 90% of Foxp3+ cells present in peripheral lymphoid organs express TNFR2 [77]. Indeed, to secrete IL-17A and recruit myeloid cells into the tumor in a mouse ovarian cancer model, CD4 T cells needs the expression of TNFR1 [121]. In addition, TNFR2 seems to be upstream of a pathway that favors Tregs proliferation within the tumor [122] leading again to promote tumor growth. If stromal cells are the main source of TNFα within the tumor, CD4 T cells also produce TNFα in sufficient amount to inhibit CD8 T cells anti-tumor response but insufficient to impact tumor cells proliferation or viability [123]. Here again, the tipping point between pro and anti-tumor TNFα effects relies on its concentration. While high concentrations will trigger tumor cell death mechanisms, low doses activate pathways that sustain tumor growth (Figure 1).

#### 3.1.6. Other Cytokines with Immunosuppressive Effect

IL-23 is a cytokine of the IL-12 family. It is an interleukin produced in response to DAMPS (Damage Associated Molecular Pattern) at the level of epithelial barriers [124]. IL-23 is present in the tumor microenvironment and triggers the production of IL-17 during Th17 differentiation in synergy with IL-6. It also contributes to the amplification of inflammation [124,125] and could be used in therapeutic strategies [79] (Figure 1 and Table 1).

Tumor cells may secrete factors to promote Th2 and TAM2 (Tumor-Associated Macrophage) polarization, which in turn amplify this type of inflammation via IL-4, IL-5 and IL-13, suppress anti-tumor Th1 polarization and responses [126], and correlate with MDSC infiltrates [83]. Nevertheless, IL-4 was recently described to have anti-tumor functions through angiogenic-dependent properties [54,81]. Besides its role in creating a macrophage-dependent tumorigenic niche, IL-33 may directly promote TGFβ elicited Tregs differentiation, suppress IFNγ, and promote Tregs stability in the tumor [84,85] (Table 1).

### 3.2. Antitumor Effects

IFNs are classified into three different types, numbered from I to III. The type I family consists of 18 members that all bind to the same heterodimer receptor composed of IFN-α/β receptor 1 and 2. IFN-α and IFN-β are the most studied type I interferons and some anticancer treatment efficacy depends on their effects on the tumor microenvironment [86]. Mainly produced by DCs within the tumor and the tumor-draining lymph nodes, these cytokines can modulate lymphocyte responses. They are notably associated with tumor-specific CD8 T cell activation [127,128,129]. Regarding CD4 T cells, IFN-α is associated with enhanced activation of CD4^+^ T cells [87] as well as a reduced frequency of Tregs within the tumor microenvironment [88] (Table 1). On the other hand, recent studies demonstrate that type I IFNs could protect cancer cells from CTLs (Cytotoxic T lymphocytes) [130]. They could be deleterious for CD4 and CD8 CAR-T (Chimeric Antigen Receptor) cells viability and it was proposed to render CAR-T insensitive to IFN I to avoid this inhibitory effect [131]. The type II family only consists of IFN-γ, which is also well studied in a cancer context [89]. Type III interferons, or interferons lambda, are the most recently discovered IFN. Structurally similar to type II IFN, their activity resembles the one of type I IFN [90]. Type III IFN are typically produced in response to viruses or bacteria and type 2 myeloid dendritic cells that have been described to be the main producer of IFN-λ [132]. IFN-λ inhibits growth and induces apoptosis of cancer cells in models of lung, liver, prostate, and breast cancer [90,91,92,93] (Table 1). T cells and NK display increased anti-tumor responses against various cancer models including melanoma, breast, and lung in the presence of IFN-λ [90]. However, this effect is most likely indirect as T cells express an extremely low amount of its receptor [133]. Its receptor, IFNLR1, is indeed expressed by few cell types and since IFN-λ has shown anti-tumor properties, it represents a great potential as an anti-cancer therapy with diminished side effects, compared to other IFN.

IL-12, derived from dendritic cells, provides an essential signal, which drives the expression of Tbet, and therefore the differentiation of Th1 effector cells [94,95,96] (Table 1). In addition, IL-2, IL-15, and IL-18 synergize with IL-12 to trigger the production of IFNγ and direct cytotoxicity of Th1 cells [67,95] (Table 1).

## 4. Effects of Diverse Treatments on CD4 T Cell Response

### 4.1. Cytokine Based Therapies

The first tests using recombinant interleukins have encountered severe toxicities problems, even leading to the death of patients. Nevertheless, many of these trials have shown immune responses during treatment, providing a proof of concept for interleukins therapy in cancer.

The first cytokine-based therapy against cancer ever administered used IL-2 but its use involves security concerns [103]. IL-2 is a pleiotropic cytokine produced mainly by T lymphocytes. It is essential for the survival of lymphocytes and plays an important role in the initiation and maintenance of antigen-specific immune responses. Promising pre-clinical evidence were obtained in several models using IL-2 based therapy with their efficacy being a consequence of T lymphocytes expansion and increase in effector functions. The high dose IL-2 regimens are very toxic and recent advances focus on modifying it to reduce toxicity and could help overcome this problem. On the other hand, low dose IL-2 regimens have been evaluated in several trials. However, in low abundance, IL-2 binds preferably to its high-affinity IL-2Rαβγ receptor expressed on Tregs leading to their induction, thus preserving an immunosuppressive milieu in tumors [134]. Thus, a major concern about IL-2 therapy is the induction of Tregs via IL-2Ra (CD25). Several variants of IL-2 with an affinity towards the other IL-2R complex expressed by cytotoxic T cells are under development [135,136,137]. Some of these have an altered IL-2Rα binding domain, to alter the induction of Tregs immunosuppressive responses while maintaining effector T cells and NK cells immune responses, like the F42A mutant [138], but it may also affect CTLs. A next-generation IL-2 is now studied to overcome all IL-2 defects: tumor targeting, toxicity, half-life, and CTLs preferential binding but not Tregs. In a recent study, Zinchen Sun and colleagues have shown the efficacy in mice of an IL-2 with several mutations to favor its affinity to CD122 and fusion with an IgG1 Fc fragment to increase its half-life and an anti-EGFR (Epidermal Growth factor-Receptor) to preferentially target the tumor [139]. This next-generation may be a new hope for IL-2 based treatments.

Years later, IL-21 became the new focus of cytokine-based therapy. IL-21 is a cytokine produced by activated CD4 T lymphocytes and its receptor is found on many populations of lymphocytes. It is known to stimulate Tfh, cytotoxic T lymphocytes, and NK cell proliferation and functions. IL-21 is essential for the differentiation of Tfh through its induction of two key transcription factors for Tfh: Bcl6 (B-cell lymphoma 6 protein) and Maf. Differentiated Tfh can subsequently produce IL-21 to stimulate B cells and the creation of a robust B cell response. IL-21 also exerts its anti-tumor effects via its stimulation of NK cells and CTL [140], two anti-tumor immune populations. However, IL-21 based therapy showed antitumor effects when associated with other treatments like checkpoint inhibitors [141] or DNA vaccines [142]. Several clinical trials using IL-21 have now been conducted in humans, alone or in combination, with the restraining aspect of a dose limitation due to toxicities (such as neutropenia or thrombocytopenia) (Figure 2).

### 4.2. Therapeutic Neutralization

Canakinumab, an IL-1β blocking antibody, is being studied as monotherapy and in combination in several types of cancer. Several trials on lung cancer, breast cancer, colorectal cancer, pancreatic cancer, renal cancer, and melanoma emphasize the interest in blocking IL-1β. For example, in 2018, a phase 3 trial of Canakinumab compared to placebo began in Non-Small Cell Lung Cancer (NCT03447769). Likewise, a phase 2 trial studying Canakinumab as monotherapy for myelodysplastic syndrome and chronic myelogenous leukemia was launched in 2020. In 2018, a phase 3 trial using Canakinumab in combination with Pembrolizumab and chemotherapy in Non-Small Cell Lung Cancer was also launched (NCT03631199) and in 2019 another one started (NCT03626545).

Anakinra is an unglycosylated form of human IL-1RA that competitively inhibits IL-1α and IL-1β from binding their receptor [143]. It has shown benefits in several clinical trials. A phase 2 trial of Anakinra in combination with dexamethasone in patients with smoldering and indolent multiple myeloma has shown increased survival of responders compared to non-responders [144,145]. Anakinra is also under investigation in several other clinical trials in pancreas carcinomas, triple-negative breast cancers, colorectal cancers, and melanoma. A phase 2 clinical study shows that the use of Anakinra restored the antitumor efficacy of 5-FU (5-FluoroUracil) in patients. Of the 32 patients enrolled, 5 presented a response (CHOI criteria) and 22 patients had a stable disease [146]. Another possible candidate is Rilonacept, the extracellular domain of IL-1RAcP (Interleukin-1 Receptor Accessory Protein) and IL-1R1 fused to the Fc part of human IgG1. It has a strong affinity with IL-1β and IL-1α, and strongly inhibits the activity of IL-1 [147]. However, these blockers inhibit both IL-1β and IL-1α, but IL-1α may have synergistic or antagonistic effects, depending on the context.

Another interleukin associated with tumorigenesis, and therefore an attractive target for immunotherapy, is IL-6. The anti-IL-6 antibody Siltuximab has been studied in cancer patients. Although it has been approved by the Food and Drug Administration (FDA) for Unicentric Castleman’s disease in 2014, its efficacy has not been proven in solid tumors. In prostate cancer and treatment of colorectal cancer, the best outcome was stable disease [148,149]. In combination with chemotherapy, the anti-IL-6R antibody Tocilizumab gave the first positive results in epithelial ovarian cancer and immunological response such as an increase in T lymphocyte activation has been observed [150] (Figure 2).

### 4.3. Treatments Targeting Regulatory T Lymphocytes

Regulatory T lymphocytes (Tregs) are an important immunosuppressive population, commonly found in tumors. One strategy to restore immune responses against a tumor is to get rid of these immunosuppressive cells. The study of the impact of chemotherapies on the immune system showed that several molecules were capable of targeting Tregs, like Cyclophosphamide [151], Paclitaxel [152], Temozolomide [153], and Imatinib [154].

Cyclophosphamide is a chemotherapy commonly used against neuroblastoma, sarcoma, and ovarian cancer amongst others. A low dose of cyclophosphamide (<300 mg/m²) is responsible for a decline in Tregs as observed in both humans and mice. The specificity of cyclophosphamide for Tregs is thought to be due to lower levels of ATP present in Tregs as compared with other T cells. This low level of ATP results in low levels of glutathione in Tregs that are consequently less efficient in detoxifying Cyclophosphamide [155]. However, this antitumor effect of Cyclophosphamide against Tregs is to be balanced by its capacity in mice and cancer patients to promote the differentiation of Th17 [156]. The role of Th17 is ambiguous, mainly depending on the cytokines present in the tumor alongside the Th17 so the promotion of its differentiation can have both positive and negative effects depending on the context [56].

Imatinib mesylate is a tyrosine kinase inhibitor of the oncogenic BCR-ABL protein (found in Philadelphia chromosome-positive chronic myeloid leukemia), KIT and platelets derived growth factor receptor-α. Imatinib has been described to inhibit Tregs activity by reducing FoxP3 expression [154]. In a model of the gastrointestinal stromal tumor, Imatinib induced Tregs apoptosis in the tumor bed by reducing the tumor expression of the immunosuppressive enzyme Indoleamine 2,3-DiOxygenase (IDO). Association with immunotherapy further improved imatinib efficiency against the tumor [157]. In patients with chronic myelogenous leukemia, imatinib-treated patients exhibited selective depletion of Tregs and a significant increase in effector/memory CD8 T cells [158].

Paclitaxel is a chemotherapeutic agent used in the treatment of various tumors like breast cancer, ovarian cancer, or lung cancer. Paclitaxel can reduce the expression of FoxP3, a master regulator of Tregs, associated with reduced inhibitory functions of these cells [159]. In patients with cervical cancer, paclitaxel was shown to induce a significant decrease in Tregs associated with increased rates of cytotoxic CD8 T cells in the tumor stroma [160].

Temozolomide is an alkylating agent known to prolong survival in patients with high-grade glioma, glioblastoma, and melanoma [161]. Temozolomide is responsible for a profound lymphopenia, including Tregs in humans but seems more specific to Tregs in mice [162] (Figure 2).

### 4.4. Immune Checkpoint Inhibitors

The identification of immune checkpoints blocking lymphocyte functions has led to the emergence of new strategies aiming at the reactivation of those lymphocytes. Immune checkpoints can be expressed by lymphocytes (PD-1, CTLA-4, TIGIT, TIM-3…) with their ligands found on tumor cells, DCs (Dendritic Cell), and APCs. Lymphocyte populations found within the tumor bed harbor higher levels of immune checkpoint expression than in their healthy counterparts and are thus less active against the tumor. Lifting the immune checkpoints dependent immunosuppression allows the return of an anti-tumor immune response. Immune checkpoint inhibitors (ICIs), consisting of blocking antibodies targeting PD-1, PD-L1, and CTLA-4, have been the first ones to be authorized as anti-cancer therapies. Pembrolizumab and Nivolumab (anti-PD-1) showed promising results in melanoma patients as well as in non-small cell lung carcinoma patients with an objective response rate of about 45% [163,164,165]. Ipilimumab (anti-CTLA-4) is used against advanced melanoma but is often responsible for immune-related side effects [166]. It is now more commonly found associated with other treatments. Atezolizumab, Durvalumab, and Avelumab are antibodies targeting PD-L1. Anti-PD-L1 antibodies are used against urothelial cancers [167], NSCLC [168], small cell lung cancer [169], kidney cancer [170], or triple-negative breast cancer [171] and show promising results, often associated with other molecules. The percentage of patients responding to ICIs varies considerably between cancers. For instance, only 19% of triple-negative breast cancer patients responded to anti-PD-1 [172] when 87% of patients with relapsed or refractory Hodgkin’s lymphoma presented an objective response to anti-PD-1 [173]. The Association of Ipilimumab and Nivolumab in treating patients is rising as observed by the many ongoing phase II and III clinical trials. These trials concern patients with advanced kidney cancer (NCT03793166, NCT04510597), advanced melanoma (NCT02339571, NCT04511013), Hodgkin lymphoma (NCT01896999, NCT02408861), glioblastoma (NCT04396860, NCT04145115), and many other cancers. Association of Ipilimumab and nivolumab has been approved by the US FDA in NSCLC patients with a PD-L1 expression ≥1% and seems to a higher survival rate in patients with PD-L1 ≥ 50% when compared to platinum-based chemotherapy but its effects on progression-free survival or overall response remains to be determined [174]. On average, however, only about 20–25% of patients respond to ICIs alone. This can probably be explained by the weak infiltration of lymphocytes within the tumor in many cancers and is the reason why ICIs are often combined with other treatments [175]. Interestingly, the tumor mutation burden is a potential predictive biomarker regarding the likelihood of a tumor to respond to ICIs [176]. Checkpoint inhibitors are now used as single agents or combined with chemotherapies in about 50 cancer types. ICIs do not target one lymphocyte specifically but rather will affect all lymphocytes bearing its target (Figure 2).

### 4.5. Chemotherapy

Gemcitabine and 5-FU are two nucleoside analogs known to suppress MDSCs in several murine tumor models [177,178]. This suppression of MDSCs results in a reduced tumor growth dependent on T cells. These effects are however transient as dying MDSCs release IL-1β that increase IL-17 producing CD4 T cells, establishing another form of immunosuppression [69]. The association of anti-IL-1β with a 5-FU treatment was able to lead to full recovery in about 40% of mice. A phase 3 clinical trial is currently ongoing in colorectal cancer patients [146]. Doxorubicin can also target MDSC [179]. In a murine mammary cancer model, doxorubicin led to the reduction of MDSC frequencies in the spleen. This depletion allowed an increase in Granzyme B and IFN-γ production by effector T cells and NK cells.

Tumor-associated macrophages can be depleted using trabectedin [180], a treatment used in the treatment of advanced soft tissue sarcoma and ovarian cancer relapses. Trabectedin depletes TAM in a range from 30 to 77% observed in clinical studies through the induction of apoptosis. A molecule similar to Trabectidin, Lurbinectedin is also capable of reducing the TAM population in the treatment of small-cell lung cancer patients, alone or in association with chemotherapies or CKI [181,182]. Other molecules, such as antibodies targeting CSF1R (Colony Stimulating Factor 1 Receptor) or liposome clodronate can also efficiently deplete TAM [183,184]. Polarizing TAM into the anti-tumor M1 phenotype is another strategy. Paclitaxel [185], Docetaxel [186], or a combination of chemotherapies (cyclophosphamide, doxorubicin, Vincristine) favor the polarization of TAM into the M1 phenotype, resulting in an increased lymphocyte response against the tumor as observed by delayed tumor growth kinetics [187].

Cytokines such as IFNγ or TNFα play crucial roles in the anti-tumor immune response. While some cancer treatments cause apoptotic cell death that is a silent cell death that does not induce an immune response, others such as anthracyclines can affect the death of cancer cells and activate the immune system. This type of cell death is called ICD (Immunogenic Cell Death). Cytokines produced during ICD may be pro-inflammatory such as TNF, IL-6, IL-8, or IL-1β. They will induce the expression of Class I MHC (Major Histocompatibility Complex) on APCs and promote T cell differentiation [188,189,190,191]. The cytokinic response induced by ICD is essential for promoting anticancer immunity involving CD4 T cells and an increase in IFNγ production by Th1 and IL-17 produced by Th17 is observed [192] (Figure 2).

### 4.6. Radiotherapy

If radiotherapy is commonly used in association with other treatments (surgery, chemotherapy, or immunotherapy more recently), it also influences the immune response on its own in tumors. Radiotherapy is responsible for an increase in Tregs [193] and it was later suggested that Tregs were more resistant to radiotherapy than other lymphocytes [194]. Radiotherapy increases tumor-infiltrated Tregs that express higher levels of CTLA-4 compared to Tregs from non-irradiated tumors [195], setting an immunosuppressive environment. The TGFβ production driven by radiation sustains the expression of FoxP3 in Tregs [195] but the association of radiation and immunotherapy against CTLA-4 resulted in long-term survival in a murine glioma model [196].

Interestingly, decreased percentages of Tregs were found in the peripheral blood of patients and a murine lung cancer model after radiotherapy [197]. A consensus is yet to be reached regarding the impact of radiotherapy on Tregs. Radiation dose, the scheme of treatment, tumor types are all variables that can influence the effect of radiotherapy on Treg and need to be taken into account (Figure 2).

## 5. Conclusions

Cytokines are key elements that orchestrate the tumor microenvironment. With a growing understanding of cancer biology and infiltrating immune cells such as CD4 T cells, the relevance of these cytokines and their functions, additional roles are likely to emerge.

Data concerning the role of a particular cytokine on CD4 T cells must be put in context, regarding the development of the tumor. Indeed, a cytokine can have an anti-tumor role at the beginning of carcinogenesis and harbor pro-tumor activity later on with the adaptation of the tumor and the impact of intra-tumor cytokines on the activity of CD4 T lymphocytes. Likewise, several cytokines exhibit antagonistic effects depending on their concentration depending on their effect on CD4 T cells among other things. These different elements must be taken into consideration before developing therapeutic strategies.

Similarly, it is important to note that the tumor environment is complex and that cytokines never evolve alone but always in concert. Thus, the effect of a therapy that would induce or conversely inhibit the effect of a cytokine can be partially prevented or even reversed depending on the cytokine milieu in the tumor microenvironment.

Cancer treatments by strategies modulating interleukins are quite complicated and require improvement to increase efficacy and decrease side effects. Recent therapeutic developments have shown that the neutralization of key pathways may have therapeutic activity in cancer patients by restoring the anti-tumor activity of CD4 T cells. For several candidates mentioned in this overview, the ongoing trials will reveal their efficacy and safety in randomized controlled settings. In line with the importance of cytokines, it is now clear that virtually all cancer treatments will modify the cytokine milieu, sometimes at the detriment of the treatment’s efficacy against cancer. Moreover, these treatments are nowadays associated with other therapies to treat patients. In particular, associations with immunotherapies such as immune checkpoint inhibitors are currently under scrutiny in various cancers, both in mice and humans. A complete understanding of the consequences of treatment on the intra-tumor cytokines would allow the establishment of more precise and efficient combinations of treatments, allowing modulation of CD4 T cells activity and increasing their effectiveness.

## Figures and Tables

**Figure 1 cancers-13-00373-f001:**
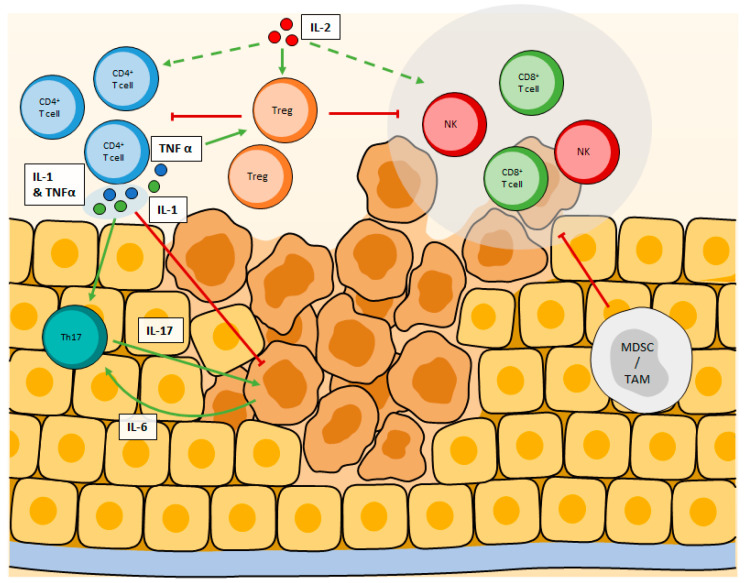
Ambivalent cytokines modulating CD4^+^ T cell responses in the tumor microenvironment. IL-2 stimulates both CD4^+^, CD8^+^, and NK cells but has a greater affinity for the IL2Rαβγ expressed by Tregs. High doses of IL-2 enhance anti-tumor immune response while low doses preferentially stimulate Tregs proliferation, leading to immune suppression. Similarly, while high doses of TNFα and IL-1 have anti-tumor effects, in the lower amount, they both stimulate Th17 cells, plus Tregs for TNFα, leading to angiogenesis and immunosuppression, which favor tumor growth.

**Figure 2 cancers-13-00373-f002:**
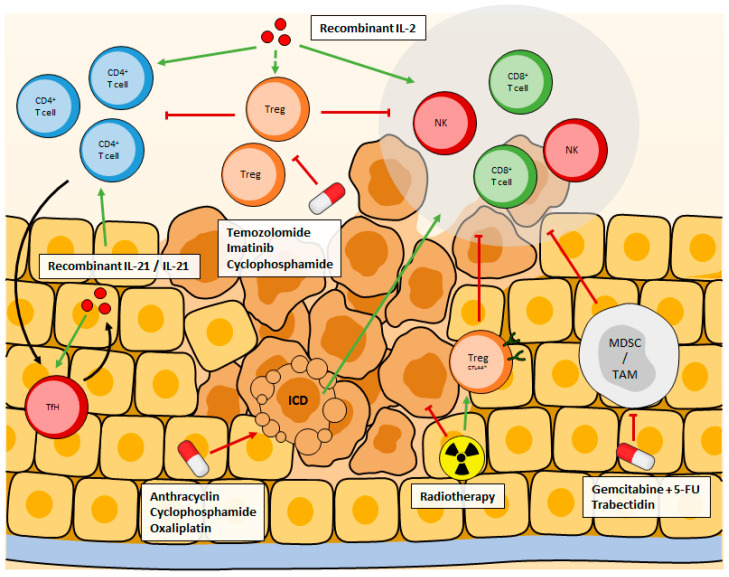
Recombinant IL-2 is designed to preferentially bind to IL2Rα or IL2Rβγ to avoid Tregs expansion, while stimulating CD4^+^, CD8^+^, and NK cells. Recombinant IL-21 also stimulates CD4^+^, CD8^+^, and NK cells as well as TFh cells, creating an auto-amplification loop. Some treatments inhibit Treg cells, MDSCs or TAM, to remove the immunosuppression they exert, while others indirectly promote anti-tumor immune response through immunogenic cell death. Radiotherapy is a double-edged sword, it directly kills tumor cells, but post-treatment it favors the recruitment of Tregs that highly express CTLA4, resulting in enhanced immune-suppression.

**Table 1 cancers-13-00373-t001:** Effects of cytokines present in the tumor microenvironment on the immune response, CD4 T cells, and tumor growth.

Cytokines	Effect on Immune Response	Associated CD4 T Cell Subsets	Effect on Tumor Growth	References
TGFβ	Immunosuppressive	Treg, Th9	Promotion	[47,50,51,56]
IL-6	Inflammatory	Th17	Promotion	[57,58]
IL-2	Inflammatory	All	Ambivalent	[59,60,61,62,63,64,65,66,67,68]
IL-1	Inflammatory	Th17, Th9	Ambivalent	[69,70,71,72,73,74]
TNFα	Inflammatory	Th1, Th17	Ambivalent	[75,76,77,78]
IL-23	Inflammatory	Th17	Ambivalent	[79]
IL-17	Inflammatory	Th17	Promotion	[80]
IL-4	Immunosuppressive	Th2, Th9	Ambivalent	[54,81,82]
IL-13	Immunosuppressive	Th2	Promotion	[83]
IL-33	Immunosuppressive	Th2	Promotion	[84,85]
IFNs type I	Inflammatory	Th1	Inhibition	[86,87,88]
IFN type II	Inflammatory	Th1	Inhibition	[89]
IFN type III	Inflammatory	Th1	Inhibition	[90,91,92,93]
IL-12	Inflammatory	Th1	Inhibition	[94,95,96]
IL-15	Immunosuppressive	Th1	Promotion	[95]
IL-18	Inflammatory	Th1	Inhibition	[95]

## Data Availability

Not applicable.

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
