# Peer review of "Modulation of CD4 T Cell Response According to Tumor Cytokine Microenvironment"

_cancers, 2021, doi:10.3390/cancers13030373_

Round 1

Reviewer 1 Report

In the manuscript entitled "Modulation of CD4 T cell response according to tumor cytokine microenvironment" the authors summarize knowledge about different immunomodulary molecules towards CD4 T cells.

I think the topic is quite interesting and the role of CD4 T cells in anti-tumor immunity so far underestimated.

With regards to this review I have several concern which should be addressed or corrected:

1) No greek symbols are displayed in my version in the text.

2) As this manuscript is a review it should "re-view" the current knowledge about the topic. Up to Reference 64 more than 60% of the references are older than 10 years. As the role of CD4 T cells is emerging, this is from my point of view far to old literature.

3) Also up to reference 64 more than 25% of the references are Reviews by themselves. I personally do not see any reason to summarize other reviews. The cited Reviews should be replaced by primary literature.

4) The main reason for a Review is to summarize current knowledge and literature of a given topic. Therefore citing of references should be done very careful.

4a) I do not see how reference 20, dealing with Bcl-6 expression, should fit to the text mentioning TGF.

4b) Reference 21, a review about Stat3 signaling, is not a suitable reference for the mentioned VEGF in the main text.

4c) Reference 41: I was not abled to find reference 41 at all. I addition from the title I guess it has nothing to do with the mains texts citation.

I do strongly recommend to carefully check the cited literature as this is the basis of a good review.

Also quite interesting: In section 5. Conclusions CD4 T cells appear with no word. This is to me quite unusual as the title is "Modulation of CD4 T cell...."

Author Response

We thank Reviewer 1 for his/her comments and we add our responds below.

Point 1) No greek symbols are displayed in my version in the text.

Response 1: We corrected these errors in the text.

Point 2) As this manuscript is a review it should "re-view" the current knowledge about the topic. Up to Reference 64 more than 60% of the references are older than 10 years. As the role of CD4 T cells is emerging, this is from my point of view far to old literature.

Response 2: We corrected this point by changing the previous references to more recent ones and implemented the new data from the literature in the text.

Point 3) Also up to reference 64 more than 25% of the references are Reviews by themselves. I personally do not see any reason to summarize other reviews. The cited Reviews should be replaced by primary literature.

Response 3: We replaced Reviews articles by original articles as soon as it was relevant.

Point 4) The main reason for a Review is to summarize current knowledge and literature of a given topic. Therefore citing of references should be done very careful.

4a) I do not see how reference 20, dealing with Bcl-6 expression, should fit to the text mentioning TGF.

4b) Reference 21, a review about Stat3 signaling, is not a suitable reference for the mentioned VEGF in the main text.

4c) Reference 41: I was not abled to find reference 41 at all. I addition from the title I guess it has nothing to do with the mains texts citation.

I do strongly recommend to carefully check the cited literature as this is the basis of a good review.

Response 4: We have checked the cited articles and corrected the errors in the text.

Point 5) Also quite interesting: In section 5. Conclusions CD4 T cells appear with no word. This is to me quite unusual as the title is "Modulation of CD4 T cell...."

Response 5: We have revised this paragraph and made corrections as shown below.

"Cytokines are key elements that orchestrate the tumor microenvironment. With a growing understanding of the cancer biology and infiltrating immune cells such as CD4 T cells, the relevance of these cytokines and their functions, additional roles are likely to emerge.

Data concerning the role of a particular cytokine on CD4 T cells must be put in context, regarding the development of the tumor. Indeed, a cytokine can have an anti-tumor role at the beginning of carcinogenesis and harbor pro tumor activity later on with the adaptation of the tumor and the impact of intra tumor cytokines on the activity of CD4 T lymphocytes. Likewise, several cytokines exhibit antagonistic effects depending on their concentration dependingon their effect on CD4 T cells among other things. These different elements must be taken into consideration before developing therapeutic strategies.

Similarly, it is important to note that the tumor environment is complex and that cytokines never evolve alone but always in concert. Thus, the effect of a therapy which would induce or conversely inhibit the effect of a cytokine can be partially prevented or even reversed depending on the cytokine milieu in the tumor microenvironment.

Cancer treatments by strategies modulating interleukins are quite complicated and require improvement to increase efficacy and decrease side effects. Recent therapeutic developments have shown that the neutralization of key pathways may have therapeutic activity in cancer patients by restoring anti-tumor activity of CD4 T cells. For several candidates mentioned in this overview, the ongoing trials will reveal their efficacy and safety in randomized controlled settings. In line with the importance of cytokines, it is now clear that virtually all cancer treatments will modify the cytokine milieu, sometimes at the detriment of the treatment’s efficacy against cancer. Moreover, these treatments are nowadays associated with other therapies to treat patients. In particular, associations with immunotherapies such as immune checkpoint inhibitors are currently under the scrutiny in various cancers, both in mice and humans. A complete understanding of the consequences of a treatment on the intratumor cytokines would allow the establishment of more precise and efficient combinations of treatments, allowing modulation of CD4 T cells activity and increasing their effectiveness."

Reviewer 2 Report

This is a review focused on the tumor cytokine microenvironment and how it impacts CD4 T cell immune responses. Information about different cytokines and their immunosuppressive and anti-tumor effects is discussed as well as the effects of different anti-cancer therapies on CD4 T cells responses. While this review incorporates an extensive literature review and presents a lot of important information to readers, in its present form it is a relatively dense piece to read, in part due to no specific order of presentation of the cytokines and/or effects on different CD4 T cell subsets. Incorporation of a Table(s) summarizing the information by cytokine and/or by CD4 T cells subset could help improve this review. Moreover, while the review is generally well-written, there are a few typos or grammatical errors that will need to be corrected. A few specific issues are:

  1. Most Greek letters for different cytokines are missing from this version of the manuscript. These need to be included.
  2. On line 52, “cancers” need to be replaced by “tumors”. It is the tumors that incorporate many cells types, not the “cancers”, which are derived from a single cell type.
  3. On line 65, “leucocytes” needs to be replaced by “leukocytes”.
  4. On line 163 the manuscript states that Tregs are the only lymphocytes to express the IL-2Ra subunit (no reference given). This is not accurate. Activated CD4 and CD8 T cells also express the high-affinity trimeric IL-2R, and thus the IL-2Ra subunit, albeit at lower expression levels than Tregs.
  5. The subtitle on line 256 refers to “Cytokine based vaccines”. However, here the authors are referring to cytokine-based THERAPIES, which are NOT VACCINES. This subtitle needs to be changed.

Author Response

We thank Reviewer 2 for his/her comments. Our answers are below.

Point 1) While this review incorporates an extensive literature review and presents a lot of important information to readers, in its present form it is a relatively dense piece to read, in part due to no specific order of presentation of the cytokines and/or effects on different CD4 T cell subsets. Incorporation of a Table(s) summarizing the information by cytokine and/or by CD4 T cells subset could help improve this review.

Response 1:To improve the manuscript, we have subdivided the part relating to the effects of cytokines on CD4 into sub-parts and added a summary table.

Point 2) Most Greek letters for different cytokines are missing from this version of the manuscript. These need to be included.

Response 2: we made the corrections in the text.

Point 3) On line 52, “cancers” need to be replaced by “tumors”. It is the tumors that incorporate many cells types, not the “cancers”, which are derived from a single cell type.

Response 3: We corrected this point in the text.

Point 4) On line 65, “leucocytes” needs to be replaced by “leukocytes”.

Response 4: We corrected this point in the text.

Point 5) On line 163 the manuscript states that Tregs are the only lymphocytes to express the IL-2Ra subunit (no reference given). This is not accurate. Activated CD4 and CD8 T cells also express the high-affinity trimeric IL-2R, and thus the IL-2Ra subunit, albeit at lower expression levels than Tregs.

Response 5: We have removed this sentence and added the sentence below with a reference.

"Tregs have the greatest affinity for IL-2 among T cells [Malek TR. 2008]."

Point 6) The subtitle on line 256 refers to “Cytokine based vaccines”. However, here the authors are referring to cytokine-based THERAPIES, which are NOT VACCINES. This subtitle needs to be changed.

Response 6: We changed "Cytokine based vaccines" for "Cytokine based therapies".

Reviewer 3 Report

In this review article, the authors described effects of cytokines on the regulation of T lymphocytes and immune profiles against tumors in the microenvironment, as well as therapeutic insight through those immunological modulations.

This article contains many abbreviations without full spelling at the first time appearance, which is not easy to understand for readers.

Checkpoint inhibitors are nowadays important options in cancer treatment. In the section 4.4, the authors described about anti-PD-1 therapy, but not about anti-PD-L1 and CTLA-4 therapy, both of which should be explained as other options of the immune checkpoint inhibition therapies. In addition, tumor mutation burden should be mentioned as a predictive factor for susceptibility for the checkpoint inhibition therapies.

Figure 2 contains IL-2, IL-21, several chemotherapeutic drugs and molecular targeted drugs, and radiotherapy, but not IL-1, IL-6 and immune checkpoint inhibitors. Can the authors include IL-1, IL-6 and immune checkpoint inhibitors in Figure 2 or Figure 3 to cover the whole image of CD4+T cell-associated cancer treatment?

In Page 9, immune checkpoint inhibitor is CKI? Originally, CKI might be cyclin-dependent kinase inhibitor.

In line 383, Page 9, “small cell cancer lung patients” should be “small cell lung cancer patients”.

In line 396, Page 9, “increase in IFN production produced among others by Th1” might be “increase in IFN production by Th1”?

Author Response

We thank Reviewer 3 for his/her comments. Our responses are below.

Point 1) This article contains many abbreviations without full spelling at the first time appearance, which is not easy to understand for readers.

Response 1: We apologize for this error and have corrected it by adding the full spellings in the text

Point 2) Checkpoint inhibitors are nowadays important options in cancer treatment. In the section 4.4, the authors described about anti-PD-1 therapy, but not about anti-PD-L1 and CTLA-4 therapy, both of which should be explained as other options of the immune checkpoint inhibition therapies. In addition, tumor mutation burden should be mentioned as a predictive factor for susceptibility for the checkpoint inhibition therapies.

Response 2: we have taken into consideration Reviewer 3's remark and modified paragraph 4.3 as indicated below.

"The identification of immune checkpoints blocking lymphocytes functions has led to the emergence of new strategies aiming at the reactivation of those lymphocytes. Immune checkpoints can be expressed by lymphocytes (PD-1, CTLA-4, TIGIT, TIM-3…) with their ligands found on tumor cells, DCs (Dendritic Cell) and APCs. Lymphocyte populations found within the tumor bed harbor higher levels of immune checkpoints expression than in their healthy counterparts and are thus less active against the tumor. Lifting the immune checkpoints dependent immunosuppression allows for the return of an anti-tumor immune response. Immune checkpoint inhibitors (ICIs), consisting of blocking antibodies targeting PD-1, PD-L1 and CTLA-4, have been the first ones to be authorized as anti-cancer therapies. Pembrolizumab and Nivolumab (anti-PD-1) showed promising results in melanoma patients as well as in non-small cell lung carcinoma patients with an objective response rate of about 45%[158-160]. Ipilimumab (anti-CTLA-4) is used against advanced melanoma but is often responsible for immune related side effects[161]. It is now more commonly found associated with other treatments. Atezolizumab, Durvalumab and Avelumab are antibodies targeting PD-L1.  Anti-PD-L1 antibodies are used against urothelial cancers[162], NSCLC[163], small cell lung cancer[164], kidney cancer[165] or triple negative breast cancer[166] and show promising results, often associated with other molecules. The percentage of patients responding to ICIs varies considerably between cancers. For instance, only 19% of triple negative breast cancer patients responded to anti-PD-1[167] when 87% of patients with relapsed or refractory Hodgkin’s lymphoma presented an objective response to anti-PD-1[168]. Association of Ipilimumab and Nivolumab in treating patients is rising as observed by the many ongoing phase II and III clinical trials. These trials concern patients with advanced kidney cancer (NCT03793166, NCT04510597), advanced melanoma (NCT02339571, NCT04511013), Hodgkin lymphoma (NCT01896999, NCT02408861), glioblastoma (NCT04396860, NCT04145115) and many other cancers. Association of Ipilimumab and nivolumab has been approved by the US FDA in NSCLC patients with a PD-L1 expression ≥1% and seems to a higher survival rate in patients with PD-L1 ≥ 50% when compared to platinum-based chemotherapy but its effects on progression free survival or overall response remains to be determined[169]. On average however, only about 20-25% of patients respond to ICIs alone. This can probably be explained by the weak infiltration of lymphocytes within the tumor in many cancers and is the reason why ICIs are often combined with other treatments[170]. Interestingly, the tumor mutation burden is a potential predictive biomarker regarding the likelihood of a tumor to respond to ICIs[171]. Checkpoint inhibitors are now used as single agents or combined with chemotherapies in about 50 cancer types. ICIs do not target one lymphocyte specifically but rather will affect all lymphocytes bearing its target. "

Point 3) Figure 2 contains IL-2, IL-21, several chemotherapeutic drugs and molecular targeted drugs, and radiotherapy, but not IL-1, IL-6 and immune checkpoint inhibitors. Can the authors include IL-1, IL-6 and immune checkpoint inhibitors in Figure 2 or Figure 3 to cover the whole image of CD4+T cell-associated cancer treatment?

Response 3: we have made the modifications suggested by Reviewer 3 and modified figure 2 (see in the manuscript please).

Point 4) In Page 9, immune checkpoint inhibitor is CKI? Originally, CKI might be cyclin-dependent kinase inhibitor.

Response 4: we apologize for this confusing error and have made the changes in the text.

Point 5) In line 383, Page 9, “small cell cancer lung patients” should be “small cell lung cancer patients”.

Response 5: we apologize for this error and have made the changes in the text.

Point 6) In line 396, Page 9, “increase in IFN production produced among others by Th1” might be “increase in IFN production by Th1”?

Response 6: we have made the changes in the text.

Round 2

Reviewer 1 Report

The authors addressed all my raised concerns satisfactorily.